# Non-trivial relationship between scaling behavior and the spatial organization of GDP in Indonesian cities

**Genta Kuno** [1]◉*, **Pradipto** [2]◉

**1** Center for Southeast Asian Studies, Kyoto University, Kyoto, Japan, **2** Yukawa Institute for Theoretical Physics, Kyoto University, Kyoto, Japan

◉ These authors contributed equally to this work.

* kuno@asafas.kyoto-u.ac.jp

**Data Availability Statement:** All data used in this study are freely available. The data for functional urban areas can be obtained from (https://ghsl.jrc.ec.europa.eu/ghs_fua.php) Data for GDP can be

## Abstract

Urban scaling analysis has shown that various aggregated urban quantities obey power-law relationships with the population size. Despite the rapid progress, direct empirical evidence that shows how the power-law exponents $\beta$ depend on the spatial organization of the GDP has been lacking. Moreover, urban scaling studies are hardly reproduced in developing countries due to inadequate official statistics. We tackle these issues by performing urban scaling analysis on Indonesian cities using globally harmonized functional cities delineations and global-scale gridded Gross Domestic Product (GDP) datasets. First, we observe that the GDP and area of Indonesian cities scale linearly with the population size. For GDP in particular, the deviations from the scaling law follow a geographical pattern. Second, we determine the economic hotspots in each city and observe that the area of the hotspots scales mildly sublinear with the population size. Surprisingly, the GDP of hotspots also scales sublinearly with the population size, indicating a lack of increasing returns due to scaling. Third, by classifying the cities based on the spatial organization of the GDP in two dimensions (heterogeneity and spatial dispersion) and examining the scaling exponents of each class, we discover a non-trivial relation between scaling behavior and the spatial organization of the GDP. Spatial dispersion strongly affects the scaling behavior in heterogeneous cities, while such effect is weakened for homogeneous cities. Finally, we find that the scaling effect in terms of economies of scale (sublinearity of area) and increasing returns (superlinearity of GDP) is stronger for Indonesian cities with spatially compact GDP distribution.

## Introduction

A comprehensive understanding regarding the fundamental process of urban system is essential to achieve sustainable urban development. The recent paradigm of viewing cities as complex systems [1] has opened the way to elucidate the intricate mechanism beneath the urban system. One of the most celebrated findings within this paradigm is the *urban scaling* framework that explains the systematic variation of urban aggregate indicators with the population

obtained from (https://doi.org/10.5061/dryad.dk1j0). Territorial boundaries can be obtained from GADM (https://gadm.org/).

**Funding:** G. K. is supported by Grant-in-Aid for JSPS Fellows Project/Area Number J23402. The funders had no role in study design, data collection and analysis, decision to publish, or preparation of the manuscript.

**Competing interests:** The authors have declared that no competing interests exist.

size in accordance with the power-law relations

$$Y = Y_0 x^{\beta}, \tag{1}$$

where $Y$ is the aggregated urban quantities, $Y_0$ is a constant prefactor, $x$ is the population of the city, and $\beta$ is the scaling exponent [2–8]. Despite its limitations [9, 10], three arguments can be made from urban scaling based on the value of $\beta$ that takes similar values for classes of quantities: (i) $\beta$ greater than 1 (superlinear scaling) for quantities driven by social interactions that experience the increasing return due to scaling such as income and crime, (ii) $\beta$ less than 1 (sublinear scaling) for quantities that are characterized by the economies of scale such as street length and area, and (iii) $\beta \approx 1$ for quantities associated with urban features that tend to be distributed evenly per capita such as housing, and water. Such power-law exponents have been recovered theoretically from micro-level interactions [11–13], as well as the fractal dimensions of road networks and population distributions [14]. Numerous studies have also been conducted to understand how the scaling exponent $\beta$ and its deviation depend on another urban indicator or classification. It has been shown that $\beta$ are affected by cities boundary [10], dragon-king effect [15], spatial hierarchy of regions [16], and commuting network between cities [17]. Others explore how deviation from the scaling law demonstrates a geographical pattern in Indian cities [5, 8]. Then, the dependence of $\beta$ for economic quantities on its internal classification has been shown in several studies such as Refs. [18–20] for income levels in Australia and United States, Ref. [21] for Gross Domestic Product (GDP) per industry types in China, and Ref. [22] for income of different occupations.

There are some attempts to find the relation between polycentricity of a city and its scaling behavior. One of the ways to quantify polycentricity is by measuring the spatial dispersion of its activity centers i.e the hotspots (the hotspots of monocentric city is less dispersed than the polycentric counterpart). Many authors assumed that such activity centers can be represented by the local maxima in the population density. Using this assumption, it has been implicitly suggested that higher polycentricity weakens the increasing return due to scaling in US cities [23]. Then, a model for cities has been proposed to demonstrate the transition from monocentric to polycentric structure [24]. The model predicts that the number of hotspots in a city scales sublinearly with the population size, which has been confirmed empirically in Spanish cities through a non-parametric hotspots detection technique based on the density of mobile phone users [25]. An implicit connection has been made through the model of Ref. [24], where the model demonstrates that the diseconomy due to congestion scales superlinearly with the population and such diseconomy is weaker in cities with higher polycentricity [26]. Alternatively, one can regard the hotspots or activity centers by using data sources other than population density that can reflect the economic activity in a more straightforward manner. Instead of using the population density data, Ref. [27] analyzed economic hotspots based on night-time light intensity in the United States, European Union, and Chinese cities. They observed an inverted U-shape curve between productivity and polycentricity i.e as cities become more monocentric, the GDP per $km^2$ increases until critical point and then decreases afterward. Then, A more straightforward approach to connect the exponent $\beta$ for GDP and its intra-urban spatial dispersion has been reported in Ref. [28], where the authors demonstrated a positive correlation between Moran's I index of the 2 $km^2$-scale GDP data and $\beta$ through numerical simulations. Yet, such correlation has not been supported by empirical observations. On top of that, measurement of spatial dispersion like Moran's I index alone cannot distinguish between polycentric structure and discontinuous development [29, 30]. As proposed in Ref. [30], one can overcome such problem by measuring the intra-urban *spatial*

*organization* defined as a measurement of both spatial dispersion (polycentricity) and heterogeneity (typically represented by the Gini index) of the urban feature of interest.

Most of the urban scaling studies have been conducted for cities in developed countries with adequate official statistics. It is concerning that only a few are done in developing countries [5, 6] since the major parts of rapidly urbanizing areas of the globe are in the developing world [31]. Also, cities in low- and middle-income countries have suffered inequality worse than cities in high-income countries [32]. To overcome this challenge, we adopt the recently published dataset of Functional Urban Areas (FUA) [33, 34], which has been used for global-scale urban scaling analysis on GDP [35]. The availability of such harmonized definition of functional cities enables us to set our locus on cities in Indonesia, a typical case of a developing country without adequate official definition of cities, while also lacking derivative socio-economic data, yet approximately 150 million people (56% of the total population) belongs to urban population [36]. In this paper, we perform urban scaling analysis in Indonesian cities for area and GDP based on gridded data with 30 arc-sec ($\approx 1\text{km}^2$ on the equator) resolution [37]. We then define the economic hotpots from gridded GDP data and discuss their scaling behavior. Finally, we classify cities based on the method of Ref. [30] that captures both the heterogeneity and spatial dispersion for gridded GDP data. With 238 Indonesian cities in the FUA dataset, we gain the merit of having a sufficient number of cities to create a classification, while still grasping the country-specific context. Moreover, the usage of a global-scale publicly available dataset ensures reproducibility of our analysis in other countries as well. More details are elaborated in Methods.

## Results

### Scaling behavior of GDP and area of Indonesian cities

We start with general scaling analysis as we plot the GDP, area, and the corresponding area of the urban center and the commuting zone against the population in Fig 1. One can see that the for all quantities, the fluctuation tends to be larger in small population regime. Such data where the fluctuation has systematic variation should not be fitted with simple linear least-squares for log-transformed quantities. Instead, the data is fitted using the log-normal distribution with free $\delta$[38] (See Methods for details). First, the GDP in Fig 1A has approximately linear scaling ($\beta = 1.004 \pm 0.072$), violating the generally consistent observation of superlinearity of economic activities in both developed [2, 4] and developing countries [6, 39].

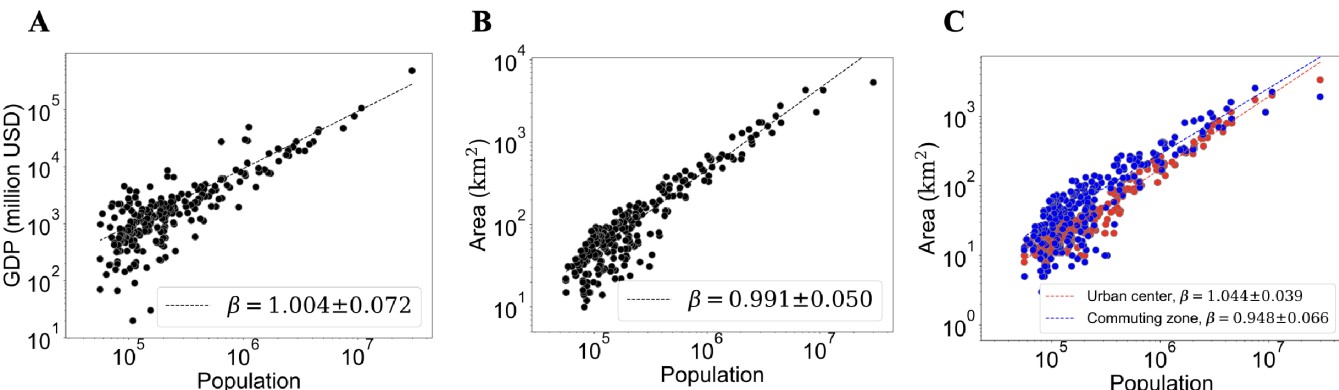

**Fig 1. Scaling law in Indonesian cities.** Plots of urban indicators of 238 Indonesian functional cities against population size. A: GDP. B: Area of the cities. C: Area of the urban center and the commuting zone. Dashed lines represent the fit from log-normal model with free $\delta$.

The linear scaling of GDP in Indonesian cities is similar to the one observed in Indian cities, where an association between a linear scaling of GDP and geographic gap in urban productivity is observed [8]. We investigate the geographic difference in GDP by defining the deviation from the scaling law $\xi_i$ as $\xi_i = \log(Y_i / Y_0 \, N^\beta)$ [5, 8]. Map of $\xi_i$ for GDP of each city can be seen in Fig 2. Then, we classify the archipelago into regions based on their timezone. Looking at the Central and East Indonesia, a dominant presence of underperforming cities can be observed with some exceptions in the eastern part of Kalimantan, which are characterized by extractive industries similar to cities in eastern Sumatra. Looking at West Indonesia, Sumatra and Java have an even distribution of underperforming ($\xi_i < 0$) and overperforming cities ($\xi_i > s0$), while the western part of Kalimantan is dominated by underperforming ones. There are several exceptional overperforming cities in the eastern part of Sumatra, where the economy of these cities is driven by oil mines and refineries. Furthermore, the magnitude of deviations tends to be lower in Java, the island with largest share of cities and population. The pattern is summarized in the violin plot of the distribution of $\xi_i$ per domestic time-zone (inset of Fig 2), as the distribution is shifting to the left (more underperforming) from west to east with a few and almost no positive deviation (overperforming cities) in East Indonesia. This observation captures the regional inequalities in Indonesia. In particular, the economic and development disparities between Java and the rest of Indonesia [40–43].

A slightly sublinear scaling for area ($\beta = 0.991 \pm 0.05$) with higher $\beta$ than the typically observed $\beta$ for area in developed countries [2, 4] is observed (Fig 1B), indicating weak economies of scale due to scaling effect. Each city consists of urban center and the corresponding commuting zone. To understand the underlying mechanism behind the weak sublinearity of the area, we examine the scaling law of urban center and commuting zone areas respectively in Fig 1C. We find a slightly superlinear scaling for urban center area ($\beta = 1.044 \pm 0.039$) and the area of commuting zone scales sublinearly with the population ($\beta = 0.948 \pm 0.066$). While the economies of scale effect due to scaling of commuting zone is as expected by the urban scaling

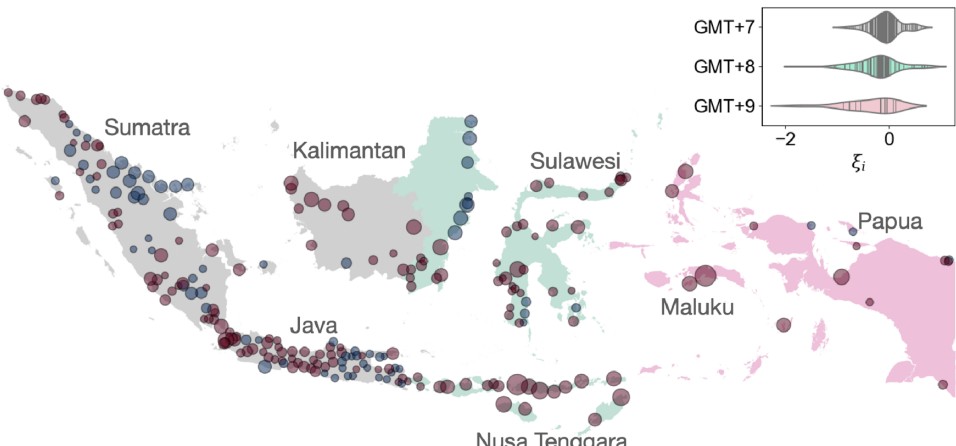

**Fig 2. Geographical pattern of GDP scaling law deviations.** Map of deviations from the GDP scaling law $\xi_i$. Red circles represent underperformance w.r.t. scaling law ($\xi_i > 0$) and blue circles represent overperformance ($\xi_i < 0$). The size of the circles represents the absolute magnitude of the deviation. The colors of islands represent different regions per time-zone (grey for GMT+7 or West Indonesia, green for GMT+8 or Central Indonesia, and pink for GMT+9 or East Indonesia). Several main islands in the archipelago are annotated as: West Indonesia (GMT+7) consisting of islands of Sumatra, Java, and the western part of Kalimantan. Central Indonesia (GMT+8) consisting of the eastern part of Kalimantan, islands of Bali, Nusa Tenggara, and Sulawesi. Then East Indonesia (GMT+9) consisting of islands of Maluku and Papua. Right inset: Violin plot of the distribution of scaling deviation grouped by the domestic time-zones. Boundaries are obtained from GADM (https://gadm.org/). Centroids of cities are obtained GHS-FUA dataset (https://ghsl.jrc.ec.europa.eu/ghs_fua.php).

theory, that of the urban center is not. Thus, it is the urban center area that disobeys the economies of scale effect due to scaling and should be clarified further.

## Scaling behavior of economic hotspots

Intra urban spatial organization of cities are characterized by the activity centers i.e the hotspots. Previous attempts define the hotspots from population density [25, 30]. However, population density does not necessarily represent the amount of activity [28]. Our approach to capture the amount of activity is by directly analyzing the gridded GDP data. While the spatial variations of the GDP data mostly driven by population density, such data that has additional variations from the subnational GDP per capita data [37]. Details on the comparison between two data sources are written in S1 Appendix. This way, the hotspots are defined as the local maxima in the surface of GDP density. From the gridded GDP data, we define the hotspots as the cells with GDP value larger than the Louail-Barthelemy (LouBar) threshold GDP* [25] (See Methods). Conversely, the non-hotspots are the cells with GDP $\leq$ GDP*. In Fig 3 the scaling behaviors of the area and GDP of both hotspots and non-hotspots are examined. For area (Fig 3A), we found that the hotspots scale sublinearly ($\beta = 0.964 \pm 0.058$) with population size, while the non-hotspots obey a superlinear scaling ($\beta = 1.045 \pm 0.066$). Note that similar to $\beta$ of urban center area, $\beta$ of hotspots area in our observation is higher than that was observed in previous research in developed countries [25, 27]. The hotspots tend to cover the core of the urban center, while the non-hotspots tend to cover parts of urban center and the commuting zone. The area of commuting zone scales sublinearly with the population (cf. Fig 1B), which implies that the non-hotspots part of the urban centers scales superlinearly with the population. Therefore, the superlinearity of the urban center area contains the contributions from the mildly sublinear area of the hotspots part and the superlinear area of the non-hotspots part.

While the non-hotspots violate the scaling effects in terms of economies of scale, Fig 3B for GDP shows that they experienced the increasing return due to scaling ($\beta = 1.072 \pm 0.055$). Such superlinearity of non-hotspots GDP indicates a transition from semi-urban to urban economics since small cities in Indonesia covers various kind of settlement entities including the peri-urban areas with mixtures of urban and rural features [44]. In contrast, the GDP of the hotspots scale sublinearly with population size ($\beta = 0.925 \pm 0.092$). This indicates that the economies of scale effect due to scaling of urban hotspots in Indonesian cities are not followed by the increasing return due to scaling counterparts.

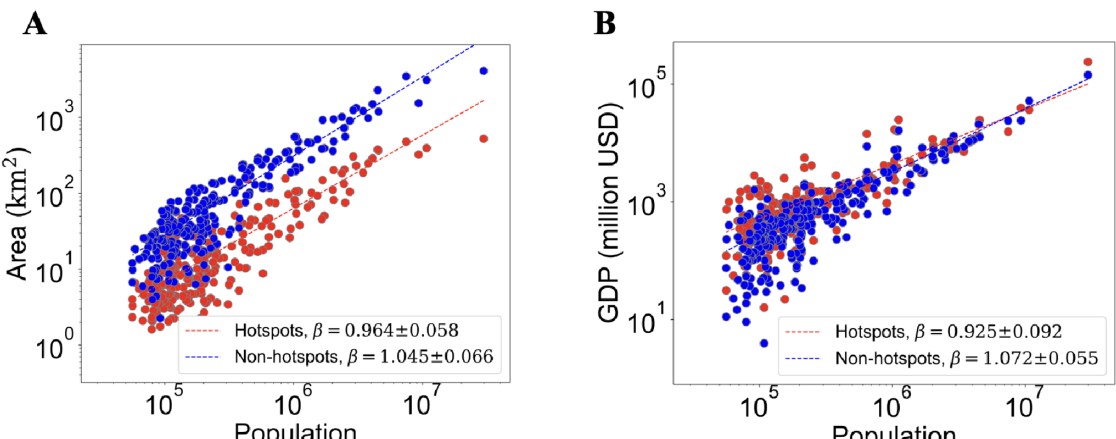

**Fig 3. Scaling law of economic hotspots in Indonesian cities.** Plots of A: GDP and B: Area of hotspots (blue) and non-hotspots (red) against population size, dashed lines represent the fit from log-normal model with free $\delta$.

## Non-trivial relations between spatial organization of the GDP and the scaling exponents $\beta$

Let us discuss the connection between spatial organization of the GDP and scaling exponents. Here we are interested in two dimensions of spatial organization of the GDP: spatial dispersion and heterogeneity. The spatial dispersion or the compactness of GDP can be captured by calculating the spreading index $\eta$, which is the ratio between the average distance of hotspots and the average distance of all cells in the city [30]. Then, the heterogeneity of the GDP distribution within cities is measured through the normalized Gini index $G_{norm}$ (See Methods for a complete description of $G_{norm}$ and $\eta$). In Fig 4A, we represent the cities in the plane ($G_{norm}$,$\eta$). We find a negative correlation between the two variables, which means that for higher GDP heterogeneity (or higher inequality), the GDP distribution becomes more compact. Consistent with past attempts [30, 45], no relation among $G_{norm}$ and $\eta$ with the population size is found. We then split the cities into quadrants by medians of $G_{norm}$ and $\eta$. This way, we have four classes of cities based on their spatial organization of the GDP: (i) 85 cities with relatively low $G_{norm}$ and high $\eta$, which are denoted as *Homo-Poly* class (green symbols in Fig 4A, top left in Fig 4B). Homo-Poly cities are associated with a homogeneous and spatially dispersed structure (more polycentric) of GDP distribution. (ii) 34 cities with high $G_{norm}$ and high $\eta$, denoted as *Hete-Poly* class (brown symbols in Fig 4A, top right in Fig 4B). Hete-Poly cities have a heterogeneous and spatially dispersed GDP distribution, which are often associated with leapfrog development [29]. (iii) 34 cities with intermediate $G_{norm}$ and low $\eta$, denoted as *Homo-Mono* class (blue symbols in Fig 4A, bottom left in Fig 4B). Cities in Homo-Mono class tend to have a monocentric structure of GDP distribution with intermediary heterogeneity. (iv) 85 cities with high $G_{norm}$ and low $\eta$, denoted as *Hete-Mono* class (purple symbols in Fig 4A, bottom right in Fig 4B). Cities that belong to the Hete-Mono class have a heterogeneous and compact GDP distribution.

Then, the scaling behavior of cities in each class is examined. From Fig 5 that shows $\beta$ of the corresponding classes, we observed a non-trivial relation between spatial organization of the GDP and the scaling exponents since each class has a distinct scaling behavior. One can see that the spatial organization of the GDP strongly affects the scaling exponents of all quantities

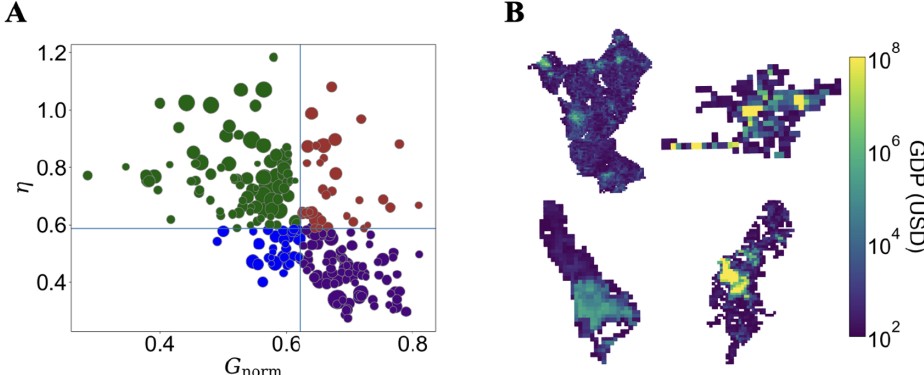

**Fig 4. Classification of Indonesian cities based on the spatial organization of the GDP.** A: Spreading index of hotspots $\eta$ versus normalized Gini coefficient $G_{norm}$, circle size represents population size, color represents the classifications. Green, brown, purple, and blue for Homo-Poly, Hete-Poly, Hete-Mono, and Homo-Mono classes, respectively. B: Spatial distributions of GDP (in 30 arc-sec grid size) of the representative cites of each class. (top left) City of Jombang, representative of Homo-Poly class, with $G_{norm} = 0.44$ and $\eta = 1.02$. (top right) City of Pamekasan, representative of the Hete-Poly Class, with $G_{norm} = 0.67$ and $\eta = 0.89$. (bottom left) City of Padang, representative of the Homo-Mono class, with $G_{norm} = 0.54$ and $\eta = 0.47$. (bottom right) City of Makassar, representative of the Hete-Mono class, with $G_{norm} = 0.71$ and $\eta = 0.36$.

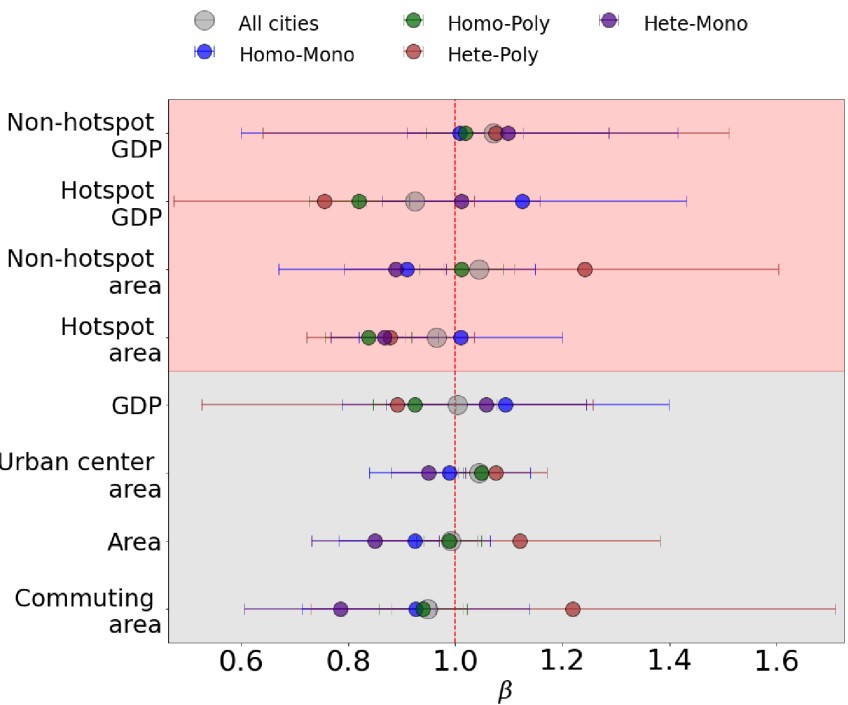

**Fig 5. Relationship between scaling exponents and spatial organization of the GDP.** Scaling exponents $\beta$ of each city class in Fig 4A, with the exponents of all cities for guidance (grey circles). Grey shaded region contains the aggregate urban quantities. Red shaded region contains urban quantities of hotspots and non-hotspots.

of interest with a couple of exceptions: (i) Scaling exponents of urban center area and hotspots area are only weakly affected since no huge differences in $\beta$, yet transition from sublinear to superlinear scaling takes place. (ii) Scaling exponents of non-hotspots GDP is almost independent of the spatial organization of the GDP since neither transition nor huge differences are observed. One can also observe that the GDP of non-hotspots is independent of its spatial organization, meaning that the semi-urban to urban transition takes place regardless of the spatial organization of the GDP. For aggregated quantities, a pattern exists such that $\beta$ for areal quantities (total area, urban center area, and commuting area) of heterogeneous cities (Hete-Poly and Hete-Mono classes) are strongly affected by $\eta$, while that of the homogeneous cities (Homo-Poly and Homo-Mono classes) are only weakly affected. Then, $\beta$ of GDP and hotspot GDP for monocentric cities are always larger than that for polycentric cities. Meanwhile, $\beta$ of commuting area, area, urban center area, and non-hotspots area of monocentric cities tend to be smaller than that for polycentric cities.

For spatially dispersed or polycentric cities (Hete-Poly and Homo-Poly classes), $\beta$ for areal quantities are always larger than that for GDP, which are the opposite of theoretical predictions [11–14] and previous observations in various countries [2–6, 8]. The aggregated and hotspots GDPs of polycentric cities scale sublinearly with populations, meaning that they experience a decreasing return due to scaling. Then, polycentric cities have weaker economies of scale effect from scaling since $\beta$ for aggregated areal quantities are always larger than the monocentric counterparts. Moreover, Hete-Poly class violates the shared trends in other classes, where $\beta$ gets gradually larger from commuting zone area, (total) area, and urban center area. On the other hand, the scaling behavior for spatially compact cities (Hete-Mono and Homo-Mono classes) obeys the general trends and theoretical predictions i.e $\beta$ for areal quantities are smaller than that for GDP. Yet, both classes have distinct hotspots scaling behavior,

where the hotspots GDP of Homo-Mono cities scales superlinearly with the population size while the hotspots area obeys a linear scaling. On the contrary, hotspots GDP of Hete-Mono cities obeys linear scaling while their hotspots area scales sublinearly with the population size. In other words, a trade-off relation exists for hotspots in monocentric Indonesian cities such that the increasing returns due to scaling comes at the expense of lacking the scaling effect in terms of economies of scale and vice versa.

## Discussions

Let us note some caveats of using the gridded GDP data. The gridded GDP data used in this paper is obtained from subnational GDP per capita and gridded population density data [37]. The resolution for subnational GDP per capita data is in general, larger than 1 km$^2$ so the spatial variation at 1 1km$^2$-scale is mostly originated from the spatial variation of the population density. Thus, such GDP data might not necessarily reflect the local economic productivity and might still represent the local population density instead.

In S1 Appendix, we repeated our analysis using population density data [46] and found that it yields same conclusions with gridded GDP data, except for the scaling exponents of the commuting area of the Hete-Poly class. When we classify the cities based on the spatial organization of the population density, the violation of the theoretical predictions observed in the Hete-Poly (gridded GDP data) is not observed. Nevertheless, further investigation is needed to compare our results with other data sources that reflect local economic productivity.

Our analysis has indicated a lack of increasing return to scale effect (sublinear scaling of GDP) of hotspots in Indonesian cities. This might be caused by two possible mechanisms: (i) As Indonesian cities grow in populations, the amount of social interaction in larger cities exceeds the optimum value so that the cost or dissipation of such interactions overcome the benefits [11]. Such non-monotonic behavior of social interactions might have a relation with the inverted U-shaped effect of hotspots compactness on economic growth [27]. (ii) There are socio-economic barriers that could not be captured through our analysis with approximately 1 km$^2$ grid size and LouBar threshold since they neither capture the GDP spread *within* hotspots nor the variation at finer resolutions, which may epitomize the Modifiable Area Unit Problem [47, 48]. Within the Indonesian context (or other developing countries), the over-cost accumulation of interaction in large cities can be associated with the so-called *urban involution* state which is often featured by proliferation of informal settlements and economies [49–52]. It has been shown that cities have distinct topological complexities at block-level, which are able to characterize informal settlements [53, 54]. One should discuss in the future how the urban scaling exponents are affected by this block-level topological complexity, since it might constrain the social interactions that are necessary to recover the superlinearity of urban economic outputs [11–13].

We have discovered a non-trivial relationship between the spatial organization (heterogeneity and spatial dispersion) of urban economy and scaling exponents in Indonesian cities. We found that the scaling behavior of aggregated quantities in heterogeneous cities strongly depends on the spatial dispersion of GDP, while homogeneous cities are weakly affected. The non-trivial relationship reveals that the scaling effect in terms of economies of scale (sublinearity of area) and increasing returns (superlinearity of GDP) is stronger for cities with spatially compact (monocentric) GDP distribution, which has been suggested earlier by Ref. [23, 28]. It can be hypothetically assumed that all social interactions that lead to superlinearity take place in the hotspots. When the hotspots are spatially adjacent, the contact probability per person is higher than the spatially non-adjacent (spreading) case. Furthermore, a polycentric city can be naively regarded as an adjacent collection of smaller cities, where each of them have their own

urban cores. Thus, when polycentric cities grow in size, the surrounding areas of each core sprawls, leading to larger physical expansions. Although monocentric cities align better with the predictions from urban scaling theory, this does not necessarily imply that monocentricity improves the quality of life. For instance, it has been shown that compared to the polycentric cities, monocentric cities suffer more from diseconomy due to congestion [9]. Also, some observed that the increasing return from scaling comes hand in hand with inequalities [19, 20]. Our results suggest that future theoretical models should consider and confirm whether a certain degree of heterogeneity and spatial dispersion is inherently required to recover the scaling effects in terms of increasing returns and economies of scale. It is also necessary to investigate whether the relationship between spatial organization of the GDP and scaling exponents $\beta$ also emerges in other countries as well to disentangle this relationship from country-specific geographical effects. Fortunately, our usage of open-source and globally harmonized definition of cities and gridded GDP datasets ensures the reproducibility of our analysis throughout the globe.

## Methods

### Data sources

Our city definition is based on the recently released GHSL-FUA dataset [34], consisting of functional urban areas (FUAs) across the globe with a harmonized definition for the year 2015. Throughout this paper, we denote FUAs as cities. Each city consists of urban center and their surrounding commuting zone. The urban center are determined based on the settlement model by GHSL-SMOD [33], and is defined as contiguous grid cells (1 $km^2$ resolution) with minimum total population of 5000, where each cell must satisfy: (i) minimum 50% built area or (ii) more than 1500 inhabitants. Then, the commuting zone is approximated based on the global friction matrix [55]. The GHSL-FUA dataset contains the population counts, boundary polygons, total areas, as well as the urban center areas of 238 cities in Indonesia. In addition, the commuting zone area is obtained by subtracting the area of the urban center from the total area of the city. We use the gridded global GDP datasets with 30 arc-sec resolution (1 $km^2$ at the equator) for the year 2015 [37]. The GDP dataset is constructed based on the national and subnational GDP data and the gridded population data [46]. Each grid in the dataset contains the GDP using purchasing power parity rates (GDP-PPP) in 2011 international US Dollars. To obtain the GDP data of each city, we overlay the gridded GDP data with city boundary polygons and sum the values at every cell. Territorial boundaries in Fig 2 are constructed from GADM database of Global Administrative Areas [56].

### Log-normal model

We estimate the scaling exponent $\beta$ using the method in Ref. [38]. Eq (1) can be rewritten as

$$\mathbb{E}(Y|x) = Y_0 x^\beta, \qquad (2)$$

with fluctuations around the model $\mathbb{V}(Y|x)$ defined as

$$\mathbb{V}(Y|x) = \mathbb{E}(Y^2|x) - \mathbb{E}(Y|x)^2 \qquad (3)$$

$\mathbb{E}(Y|x)$ is the expectation of $Y$ with respect to $x$. Here, we are interested in the case where $\mathbb{V}(Y|x)$ also scales with population size such as

$$\mathbb{V}(Y|x) = \gamma \mathbb{E}(Y|x)^\delta, \qquad (4)$$

with $\gamma$ is constant prefactor and $\delta \in [1, 2]$. The best fit then can be obtained by minimizing the

negative of log-likelihood $\ln\mathcal{L}$ with log-normal distribution

$$\ln\mathcal{L} = \sum_{i=1}^{N} -\ln(\sigma(x_i)\sqrt{2\pi}) - \ln Y_i - \frac{(\ln Y_i - \mu(x_i))^2}{2\sigma^2(x_i)}. \qquad (5)$$

$\mu(x)$ and $\sigma^2(x)$ are the mean and variance of the log-normal distribution and can be calculated from scaling parameters [38]

$$\mu(x) \quad = \ln Y_0 + \beta \ln x - \frac{1}{2}\sigma^2(x), \qquad (6)$$

$$\sigma^2(x) \quad = \ln[1 + \gamma(Y_0 x^\beta)^{\delta-2}]. \qquad (7)$$

The minimization is done using L-BFGS-B algorithm [57] in SciPy [58] with random initial parameters for 20 realizations. Finally, the errors are computed using bootstrap. All of these procedures are done using the publicly available code [59].

## LouBar threshold, normalized Gini index, and spreading index

The LouBar threshold GDP* is determined using non-parametric method as follows. For any quantity $Q$ with corresponding Lorenz curve $L(F)$, $Q^*$ is the value of $Q$ that corresponds to $F^*$, where $F^*$ is the intersection of the $x$−axis ($L = 0$) with the tangent of the Lorenz curve at $F = 1$. The LouBar threshold gives us a more restrictive criterion of a hotspot than the typical naive average value $\bar{Q}$ as a threshold since it also depends on the dispersion of $Q$ ($Q^*$ increase as the maximum value of $Q$ increases). Thus, a grid is classified as a hotspot if the GDP in the grid satisfies $GDP > GDP^*$. After such hotspots are determined, we sum their GDP to get the total GDP of hotspots. Then, as in previous studies [25, 27], we also obtain the number of hotspot grids in each city, which is treated as the total area of hotspots in a city considering that one grid has the dimension of approximately 1 km$^2$.

The heterogeneity of the GDP distribution within cities with total number of cells $N_g$ is measured through the normalized Gini index $G_{norm}$, defined as $G_{norm} = G/G_{max}$. $G$ is the classical Gini index and $G_{max}$ is the maximum Gini index of each city, defined as $G_{max} = (N_g - 1)/Ng$ i.e the maximum value of $G$ when all of the GDP are concentrated in one cell [30]. The spatial organization of hotspots can be captured by calculating the spreading index $\eta$, which is the ratio between the average distance of those hotspots and the average distance of all cells in the city,

$$\eta = \frac{\frac{1}{N_{g,GDP^*}}\sum_{i,j}d(i,j)\Theta(GDP_i - GDP^*)\Theta(GDP_j - GDP^*)}{\frac{1}{N_g}\sum_{i,j}d(i,j)}, \qquad (8)$$

where $\Theta(x)$ is the Heaviside step function. For cells $i$ and $j$, $GDP_i$ and $GDP_j$ represent the GDP value on cells $i$ and $j$, respectively. $N_{g,GDP^*}$ is the number of cells with GDP larger than GDP*, and $d(i, j)$ is the distance between cells $i$ and $j$.

## Supporting information

**S1 Appendix. Relation between spatial organization of the population and the scaling exponents $\beta$.** Contains all the supporting figures.
(PDF)

## Author Contributions

**Conceptualization:** Genta Kuno, Pradipto.

**Data curation:** Genta Kuno, Pradipto.

**Formal analysis:** Genta Kuno, Pradipto.

**Funding acquisition:** Genta Kuno.

**Investigation:** Genta Kuno, Pradipto.

**Methodology:** Genta Kuno, Pradipto.

**Project administration:** Genta Kuno, Pradipto.

**Resources:** Genta Kuno, Pradipto.

**Software:** Genta Kuno, Pradipto.

**Validation:** Genta Kuno, Pradipto.

**Visualization:** Genta Kuno, Pradipto.

**Writing – original draft:** Genta Kuno, Pradipto.

**Writing – review & editing:** Genta Kuno, Pradipto.

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
