## [Decision Letter · Decision Letter 0]

21 Jul 2022

PONE-D-22-06388Non-trivial relationship between scaling behavior and the GDP microstructure in Indonesian citiesPLOS ONE

Dear Dr. Kuno,

Thank you for submitting your manuscript to PLOS ONE. After careful consideration, we feel that it has merit but does not fully meet PLOS ONE’s publication criteria as it currently stands. Therefore, we invite you to submit a revised version of the manuscript that addresses the points raised during the review process.

Given the reviewer’s comments, what this manuscript needs to be publishable is

more description about the limitations related to the data and the degree to which you are sure that the data represents the product of local economic activity

discussion around how any uncertainty may affect your conclusions not just with respect to scaling, for example if not only represents some aspects of local income but with a bias reflecting its estimation.

We look forward to receiving your revised manuscript.

Kind regards,

Judi Hewitt

Academic Editor

PLOS ONE

Journal Requirements:

G. K. is supported by Grant-in-Aid for JSPS Fellows Project/Area Number 20J23402.

G. K is supported by Grant-in-Aid for JSPS Fellows Project/Area Number 20J23402.

However, funding information should not appear in the Acknowledgments section or other areas of your manuscript. We will only publish funding information present in the Funding Statement section of the online submission form. 

G. K. is supported by Grant-in-Aid for JSPS Fellows Project/Area Number 20J23402.

4. We note that Figure 2 in your submission contain [map/satellite] images which may be copyrighted. All PLOS content is published under the Creative Commons Attribution License (CC BY 4.0), which means that the manuscript, images, and Supporting Information files will be freely available online, and any third party is permitted to access, download, copy, distribute, and use these materials in any way, even commercially, with proper attribution. For these reasons, we cannot publish previously copyrighted maps or satellite images created using proprietary data, such as Google software (Google Maps, Street View, and Earth). For more information, see our copyright guidelines: http://journals.plos.org/plosone/s/licenses-and-copyright.

Reviewers' comments:

Reviewer's Responses to Questions

**Comments to the Author**

1. Is the manuscript technically sound, and do the data support the conclusions?

Reviewer #1: No

2. Has the statistical analysis been performed appropriately and rigorously? 

Reviewer #1: Yes

3. Have the authors made all data underlying the findings in their manuscript fully available?

Reviewer #1: Yes

4. Is the manuscript presented in an intelligible fashion and written in standard English?

Reviewer #1: Yes

5. Review Comments to the Author

Reviewer #1: This paper attempts at establishing relationships between “urban productivity” and the degree of polycentricity in Indonesian urban areas through measuring scaling effects. The text is pleasant to read, the data processing seems to be clean, but I am afraid that its contribution to existing literature remains marginal, not because of the author’s work that seems sound enough but because of the uncertainties linked with the data that are used. The text is clearly written and methodology and results are well explained. Illustrations are useful, well designed and clearly identified. Sources are clearly presented but without enough words about their limitations, especially regarding GDP measurement significance and allocation in the spatial grid. Moreover, because of the methodology that was used for building the data base, it is not at all certain that the data that are provided do represent the product of local economic activity.

To be clear: naming “GDP microstructure” the simple statistical spatial pattern that correspond to the source of data (which is a disaggregation of broad GDP figures according to the population) could be misleading because the term structure may lead to imagine information about the real networks and value chains among the economic stakeholders inside a city.

Second, it is not sure that GDP from the data base do measure urban productivity, especially at intra-urban scale, it may represent local income but also reflect the general hypothesis that were used for its estimation.

On the theoretical level, I find in the paper a few remarks reminding of the former attempts made decades ago to adapt central place theory that was conceived to describe interurban patterns, to the internal structure of cities. Intra-urban space has a different structure because the socio-spatial processes that differentiate the parts of a city (i.e. within a daily urban system) are not the same as those shaping the functional and spatial differences between more distant cities. Scaling laws were theoretically built for representing interurban relationships, their validity is less well established inside urban areas.

Some conclusions are probably going beyond what the data authorize according to the many uncertainties and weaknesses from the source that do not always represent accurately what is expected. For instance, a sentence like “Regardless of their heterogeneity, hotspots in monocentric cities in Indonesia experience the physical expansions of hotspots that are not followed by the economic growth” should be highly hypothetic. As well the first paragraph of the discussion raises questions without taking care of the real significance of the data.

Details

line 14 The expression « human needs » is not accurate for social scientists. Needs are so highly depending on the wealth and culture of societies, they have so much evolved over time that they have to be considered no longer as human but always socially defined, varying and evolving. According to regional inequalities in Indonesia, it may be that the computed scaling relationship between population and wealth reflect quite heterogeneous local economic situations.

lines 27-30 “spatial microstructures of cities “should be defined here. Does it have to do with the physical and socio-economic morphology of urban areas? What is the difference with the polycentric structures? Why using a different vocabulary ?

lines 99-101 very unclear sentence: “This implies that the suppressed fluctuation in higher population regime (Fig. 1) is coupled with the geographical constraint”. What do you mean?

lines 106-116 take for granted that the sublinear behavior of urban surface with population guaranties “efficiency of growth”. This is a biased interpretation of urban spatial structure because it do not take into account the centre-periphery process.

6. PLOS authors have the option to publish the peer review history of their article (what does this mean?). If published, this will include your full peer review and any attached files.

Reviewer #1: No

---

## [Author Response · Author response to Decision Letter 0]

28 Sep 2022

Thank you for reviewing our work. The response letter is attached.

---

## [Editor Report · Decision Letter 1]

27 Oct 2022

Non-trivial relationship between scaling behavior and the spatial organization of GDP in Indonesian cities

PONE-D-22-06388R1

Dear Dr. Kuno,

We’re pleased to inform you that your manuscript has been judged scientifically suitable for publication and will be formally accepted for publication once it meets all outstanding technical requirements.

Kind regards,

Judi Hewitt

Academic Editor

PLOS ONE